# Macro- and Microscopic Characterization of Components of Resistance against *Puccinia striiformis* f. sp. *tritici* in a Collection of Spanish Bread Wheat Cultivars

Rafael Porras [ID], Cristina Miguel-Rojas *[ID], Alejandro Pérez-de-Luque [ID] and Josefina C. Sillero [ID]

IFAPA Alameda del Obispo, Area of Genomic and Biotechnology, Avda. Menéndez Pidal s/n, 14004 Cordoba, Spain; jrafael.porras@juntadeandalucia.es (R.P.); alejandro.perez.luque@juntadeandalucia.es (A.P.-d.-L.); josefinac.sillero@juntadeandalucia.es (J.C.S.)
* Correspondence: cristinademiguelrojas@gmail.com

**Abstract:** Yellow (stripe) rust, caused by the biotrophic fungus *Puccinia striiformis* f. sp. *tritici* (*Pst*), stands as the most serious wheat disease worldwide, affecting approximately 88% of world wheat production. Even though yellow rust generally develops in cool humid weather conditions, the expansion of new races adapted to warmer climates threatens zones where severe *P. striiformis* epidemics were infrequent, such as Andalusian wheat cropping areas. In order to characterize yellow rust resistance mechanisms in Spanish germplasm, our study evaluated 19 Spanish bread wheat cultivars against *P. striiformis* under controlled conditions for percentage of disease severity (DS) and infection type (IT). From this visual evaluation, 74% of evaluated cultivars showed resistant responses against *P. striiformis* infection with only five cultivars considered susceptible. Subsequently, macroscopic and microscopic components of resistance were identified through image analysis and histological studies, respectively, in six selected cultivars. Macroscopic parameters such as total pustule area and total affected area (%), together with microscopic parameters such as early-aborted and established microcolonies regarding plant cell death responses (%), and microcolony length (μm), were identified as capable of differentiating cultivars quantitatively. Thus, these parameters could be used as a basis for screening resistant responses in future breeding programs, complementary to physiology, genetic and biochemical studies of plant-*Pst* interaction. Finally, our study seems to be the first macroscopic and microscopic characterization of *P. striiformis* infection in a collection of Spanish bread wheat cultivars in controlled conditions.

**Keywords:** yellow (stripe) rust; plant breeding; plant resistance; foliar disease; *Triticum aestivum*

## 1. Introduction

Wheat is considered the most widespread crop in the world, accounting for 760 million tons of production on 219 million hectares globally in 2020 and being of great importance in human diets [1]. This crop cultivation is divided into bread (common) wheat (*Triticum aestivum* L.) and durum wheat (*Triticum turgidum* L. subsp. *durum*),which are both expected to increase in consumption by up to 60% by 2050 due to human population growth [2]. However, this requirement for wheat production could be hard to achieve due to diverse abiotic and biotic stresses, especially fungal diseases, which cause more than 21% of wheat yield losses on average [3]. In particular, cereal rusts are considered among the most damaging diseases, causing important yield losses globally [4]. One of them, the wheat yellow (stripe) rust, caused by the biotrophic fungus *Puccinia striiformis* Westend f. sp. *tritici* (*Pst*), stands as the most serious wheat disease worldwide [5]. It was present in more than 60 countries in 2000 to 2009 with diverse cropping systems, growing seasons and germplasm characteristics [6]. This disease has undergone important global expansion in the last 50 years, and it is believed to affect approximately 88% of the world's wheat production, causing losses of 5.5 million tons per year [7].

*P. striiformis* is a biotrophic, macrocyclic and heteroecious fungus that depends on a living primary host (wheat or grasses) and an alternate host (*Berberis* or *Mahonia* spp.) for growth and reproduction [8]. The infection process on cereals begins via an urediniospore germ tube, which penetrates the stoma without differentiating an appressorium (unlike other rust fungi), and then differentiates into a substomatal vesicle (SSV) within the stomatal cavity [9]. This vesicle forms three primary infection hyphae that induce the development of haustorial mother cells (HMC) at the point of contact with host mesophyll or epidermal cells, then penetrating the host cell walls and forming haustoria by invaginating the plasma membrane [10]. The secondary infection occurs via development of intercellular runner hyphae throughout the leaf, thus developing an extensive hyphal network, producing further haustoria [11]. Finally, at the same time as the secondary infection develops, a pustule bed is established, from which an uredinium develops. The plant responses commonly appear from six to eight days after infection in the form of chlorosis, whereas sporulation (characteristic yellow-orange uredinia appearing in long, narrow stripes on leaves) starts approximately from 12 to 14 days under favorable conditions [8], leading to desiccation of leaves.

The most favorable conditions for large-scale yellow rust epidemics are temperate regions with cool humid climates [5]; therefore, the disease is considered a low-temperature disease. Thus, *P. striiformis* urediniospores germinate rapidly in optimum conditions of dew surfaces and a temperature range between 7 and 12 °C, with ideal disease development conditions (from infection to sporulation) between 12–15 °C, which taken together, are temperature ranges approximately 10 °C lower on average than those for other wheat rusts [12]. Despite these optimum conditions for yellow rust development, in the past two decades new races adapted to warmer climates have emerged, spreading into new territories and producing epidemics in warmer areas where the disease was previously infrequent or absent [8]. This situation, together with its long-distance dissemination capability, its high rates of mutation from avirulence to virulence, and the existence of recombinant and highly diverse populations [13], is leading to a control cost for the disease of more than 1 billion USD annually [5]. For that reason, the control of this disease through resistance breeding strategies stands as an efficient, economical and environmentally friendly method to prevent epidemics in the short and long-term future.

The genetic control of wheat yellow rust was achieved thanks to the steady work of plant breeders and pathologists on the identification of yellow rust (*Yr*) resistance (*R*) genes over the last 100 years [14]. In general, *R* genes in plants encode nucleotide-binding site leucine-rich repeat proteins that recognize specific pathogen effectors (avirulence proteins) [15]. Usually, the *Pst*–wheat interaction follows a classical gene-for-gene model. Recognition between *R* genes and avirulence (*Avr*) genes may lead to effector-triggered immunity, resulting in a resistant phenotype (incompatible interaction). Additionally, when effectors do not recognize *Avr* genes, a susceptible (compatible interaction) phenotype may evolve [16]. Two major types of resistance, all-stage resistance (ASR) and adult-plant resistance (APR), have been characterized and used in breeding programs for *Pst*-resistant cultivars in wheat. ASR is effective through all growth stages, being characterized by a strong to moderate immune response that restricts pathogen development, sporulation and infection. This kind of resistance is qualitatively inherited and easily incorporated into adapted cultivars. However, since it counters only one or a few *P. striiformis* races, new virulent races rapidly overcome it and varieties become susceptible after a few years [15,17,18]. On the other hand, APR is non-race specific, working against more than one race of the pathogen, a durable type of resistance due to its polygenic nature [19]. Specifically, APR genes are described as slow-rusting genes, leading to delayed infection and spore germination [20]. Currently, the most powerful strategy to combat wheat yellow rust is the combination of classical qualitative *R* genes and nonclassical quantitative *R* genes.

The *P. striiformis* clonal population present in Europe has largely been replaced since 2011 by diverse new lineages known as Warrior, Kranich and Warrior(–), causing increased epidemics on multiple wheat varieties according to various studies [13,21–23]. Currently,

according to the Global Rust Reference Center (GRRC), PstS10 is the most prevalent genetic group on bread wheat, dominated by a single race, but the continuous appearance of new races by 2020 resulted in quantitative shifts in rust susceptibility in some local wheat varieties, overcoming long-standing resistance genes [24]. In Europe, Spain stands among the main bread wheat cultivation producers, with 7.3 million tons of yield in 1.6 million hectares [25]. Thus, yellow rust disease, along with its new races, presents a serious threat to Spanish bread wheat cropping areas. At first, yellow rust epidemics generally occurred locally or regionally, and only occasionally became severe [26], possibly due to the use of varieties with genes resistant to the *P. striiformis* races present in that moment, together with the absence of optimum conditions for yellow rust development. However, the appearance of new races adapted to warmer climates and with higher aggressiveness [27] is leading a new perspective on managing this disease in Spain. In fact, yellow rust was first found in 2008 and 2009 in Northern Spain [28], but quickly spread to the Southern and Eastern regions, where severe epidemics of *P. striiformis* occurred in areas of Castilla y Leon, Aragon and Andalusia by 2013 and 2014 [29]. In 2018, a new yellow rust race named PstS14, which is virulent across various resistance genes, was detected in Spain [28]. To date, the GRRC has cataloged the Spanish isolates in the Warrior(–) (with genetic group PstS10) and PstS14 races.

In this context of possible yellow rust epidemic threats in Spanish wheat cropping areas, it appears necessary to conduct a detailed study of the infection process of *P. striiformis* in Spanish bread wheat varieties under controlled conditions, thus obtaining a complete assessment. Hence, different evaluation methods should be considered, both macroscopic and microscopic, to identify components of resistance that might help to find diverse sources of resistance to *P. striiformis* and determine the most effective, economical and environmentally friendly method to control yellow rust [30]. The most common method to evaluate this disease is through visual assessment, a subjective evaluation which generally measures parameters such as disease severity (DS, percentage of infected leaf area) or infection type (IT, index of plant–pathogen interactions determined by the proportion of host response versus fungus sporulation) [5]. However, in recent years some studies developed assessments for plant diseases through diverse image analysis systems, leading to important progress [31–33]. These image-based analysis systems allowed for the exploration of more dimensions of disease phenotypes, such as quantification of fungal reproduction structures [34], distinction among genotypes with different levels of disease severity [35], or improved reproducibility and sensitivity in disease quantification [36]. Moreover, the development of diverse software for image analysis permitted a high degree of automation in crop disease assessments [37,38] and, therefore, the possibility of developing specific applications adapted to different crop diseases [39,40]. Concretely, some studies developed methods for the macroscopic evaluation of *P. striiformis* through image analysis, which led to the identification of resistant responses among different cultivars or even the aggressiveness of diverse *P. striiformis* races, evaluating parameters such as IT, latency period (LP), infection frequency (IF), pustule size (PS), disease area (DA), etc. [41–43]. In addition to this, microscopic evaluation of plant–pathogen interaction is commonly studied through histopathological methods, investigating morphological changes in plants and pathogens at the cell and tissue levels using light, electron or fluorescence microscopy [44]. These methods provide valuable information regarding the pathogen infection process, such as the tracking of its development through the host tissue, the identification and quantification of fungal infection structures, and the recognition of host defense responses in the form of hypersensitive response (HR) in the host cells or the deposition of lignin and callose compounds [9,11,45–47], as well as the screening of resistance genes based on wheat phenotype [48].

In summary, taking into consideration that macroscopic and microscopic methods of yellow rust evaluation are valuable tools for finding sources of resistance to *P. striiformis*, together with the incoming threat of new races adapted to warmer climates in Spain, it is appropriate to conduct a specific assessment of yellow rust infection in Spanish bread

wheat cultivars. For that reason, the main objective of this study was to evaluate a group of Spanish bread wheat cultivars against a local isolate of *P. striiformis* under controlled climatic conditions through both macroscopic and microscopic methods, in order to determine the sources and mechanisms of resistance available among the studied accessions.

## 2. Materials and Methods

### 2.1. Plant Material

In our study we evaluated 19 bread wheat (*Triticum aestivum* L.) accessions against a local isolate of yellow rust (*Puccinia striiformis* f. sp. *tritici*) collected from Santaella (Córdoba, Spain) in 2020 with a virulence/avirulence spectrum as follows: *Yr1*, *Yr2*, *Yr3*, *Yr6*, *Yr7*, *Yr9*, *Yr17*, *Yr18*, *Yr25*, *Yr27*, *Yr32*, *Sp*, *AvS* / *Yr4*, *Yr5*, *Yr8*, *Yr10*, *Yr15*, *Yr24*, *Yr26*, *Yr33*, *YrAmb*. All accessions studied were commercial Spanish cultivars, registered in the Spanish MAPA catalogue (see Supplementary Table S1), either recently registered or widely cultivated by Spanish farmers. After preliminary disease screening, six wheat accessions with diverse and representative infection types (ITs) were selected for microscopic and macroscopic assays of components of resistance. These accessions were: 'Rota', 'Galera', 'Artur Nick', 'Ecodesal', 'Califa Sur' and 'Nogal'.

### 2.2. Pathogen Isolation

*P. striiformis* f. sp. *tritici* was isolated from a naturally infected field of Santaella, Córdoba (Spain). Spores from naturally infected leaves were placed onto uninfected plants of susceptible cultivar 'Califa Sur' in order to purify the inoculum. Plants were then misted with distilled water without run-off before being sealed in dark plastic bags to provide 100% relative humidity (RH), and kept for 24 h in a cool chamber at 8 °C. After 24 h, plastic bags were removed and plants were transferred to an incubator at 17/13 °C day/night, with 70% RH and a 16 h photoperiod, for 15 days. When individual pustules appeared, a single-pustule isolate was obtained by inoculation of new uninfected 'Califa Sur' plants. In order to increase the amount of spores for further inocula, 14-day-old 'Califa Sur' plants were inoculated with spores mixed with talc (1:20 *v/v*) using a manual airbrush spray, and incubated as described above. Spores of yellow rust were collected from infected leaves using a vacuum bomb, placed in a desiccator at 4 °C for 4 or 5 days, and then stored at −80 °C until inoculation experiments.

### 2.3. Screening of Bread Wheat Germplasm

#### 2.3.1. Inoculation Assays for Evaluation of Disease Severity (DS) and Infection Type (IT)

Seeds of 19 bread wheat accessions were sown in $8 \times 7 \times 7$ cm pots containing a mix (1:1, *v/v*) of commercial compost (Suliflor SF1 substrate; Suliflor, Radviliškis, Lithuania) and sand. Pots were placed in trays and incubated in a growth chamber at 21 °C with a 14 h photoperiod for germination. After seven days, seedlings were transferred to an incubator at 17/13 °C day/night, 70% RH, and 16 h of light to acclimate the plants during five days until the second leaf was halfway unfolded. Then, six seedlings of each accession were inoculated with yellow rust spores mixed with talc (1:20 wt/wt) using a manual airbrush spray, as described in Section 2.2. A total of 114 plants were uniformly inoculated with 20 mg of yellow rust spores. The experiment was performed three times. The same analysis was performed twice with adult plants (two months) to confirm the absence of variation in resistance or susceptibility compared to seedlings (Supplementary Figure S1).

#### 2.3.2. Disease Assessment via DS and IT

The second leaf of each plant was evaluated 15 days post-inoculation for seedling plants, and the fifth leaf for adult plants. The infection process was quantitatively scored as the percentage of each leaf with disease symptoms (pustules, chlorosis and necrosis), referred to as disease severity (DS). In addition, seedling reactions were qualitatively scored using a disease scoring scale (0–9) for infection type (IT) [49], where 0 = no visible disease symptoms (immune), 1 = minor chlorotic and necrotic flecks, 2 = chlorotic and necrotic

flecks without sporulation, 3–4 = chlorotic and necrotic areas with limited sporulation, 5–6 = chlorotic and necrotic areas with moderate sporulation, 7 = abundant sporulation with moderate chlorosis, 8–9 = abundant and dense sporulation without notable chlorosis and necrosis. Infection types 0–6 were considered resistant, while types 7–9 were considered susceptible.

### 2.4. Characterization of the Response to Yellow Rust

2.4.1. Inoculation Assays for Evaluation of Macroscopic and Microscopic Resistance Components

Seeds of the six selected bread wheat accessions were sown in $30 \times 20 \times 7$ cm trays containing the mix of commercial compost and sand (1:1, *v/v*) described in Section 2.3.1. Trays were incubated at 21 °C with a 14 h photoperiod to germinate and grow the plants. After 16 days, seedings were transferred to an incubator at 17/13 °C day/night, 70% RH, and 16 h light to acclimate the plants for five days until the third leaf was completely unfolded. Following Sørensen et al. [43] with minor modifications, the third leaves of four plants per accession were fixed horizontally (adaxial surface up) on a foam board with metal clips. A total of 24 leaves were fixed per tray. Each tray was inoculated with 4 mg of yellow rust spores mixed with talc (1:20 wt/wt) using a settling tower to inoculate the leaves uniformly. The inoculation assay was continued as described in the pathogen isolation section. Microscopic experiments were performed three times while macroscopic experiments were performed two times.

In order to study the progression of the disease *in planta* through the leaves of the six selected accessions, an additional tray with fixed leaves was inoculated as described above. Before inoculation, leaves were fully covered with a plastic lid except for a 1 cm length section in the middle of the leaves, where spores settled. After inoculation, the plastic lid was removed, and plants were kept in the same condition as previously described for 15–19 days. This experiment was performed two times.

2.4.2. Assessment of Macroscopic Resistance Components

The procedure followed was similar to that in Sørensen et al. [42] with minor modifications. Third leaves of each plant, fixed on cork pedestals, were detached 15 days post inoculation, placed on sheets of black cardboard (A4), and digitally scanned (Canon CanoScan LiDE 400, Tokyo, Japan) at 1200 ppi resolution. The image analysis software Fiji [38] was used for analyzing the following parameters: infection frequency (IF, number of pustules per $cm^2$ of leaf), mean pustule size ($cm^2$), total pustule area relative to leaf area (%), total affected area (chlorosis and necrosis) relative to leaf area (%), and proportion of pustule area relative to total disease area (pustule area plus affected area). Image analysis was conducted on 6 $cm^2$ of individual leaves, four leaves per accession. Areas of pustules and disease symptoms were determined by the color thresholding option using the default method with the HSB color space setting.

In addition, the third leaves covered before inoculation except for a 1 cm length area were studied at different time intervals to determine *P. striiformis*'s latency period (LP0) and final lesion length [42]. Evaluation of LP0 started eight days after inoculation with the examination of inoculation sites with a hand lens at 24 h intervals for 19 days. LP0 was defined as the time interval (hours) from inoculation until the first appearance of spores in the uredinia breaking the leaf epidermis. Evaluation of lesion growth started at the same time as LP0 for individual leaves and was carried out by marking the expanding edge of the lesion using a waterproof felt-tip pen (Staedtler, Nuremberg, Germany). This marking was repeated at 4-day time intervals for three consecutive measurements. Once final markings were carried out, leaves were detached and digitally scanned as previously mentioned. Total length of each lesion was evaluated using Fiji software [38].

2.4.3. Assessment of Microscopic Resistance Components

The central leaf segments (approximately 6 cm) of third leaves of plants were cut seven days post-inoculation to evaluate the fungal development stages in different accessions. Samples were processed as in Soleiman et al. [46] with minor modifications. Briefly, leaves were fixed and cleared through boiling for 5 min in lactophenol/ethanol (1:2), and stored overnight at room temperature. Samples were then transferred into a saturated chloral hydrate solution (5:2 wt/v) for 24 h. Then, they were washed once with 50% ethanol for 30 min, twice with 0.5 M NaOH for 15 min each, rinsed three times in distilled water (10 min each), and soaked in 0.1 M Tris buffer (pH 8.5) for 30 min. Leaf segments were stained with Uvitex 2B fluorescent dye (Polysciences, Bergstrasse, Germany) using a 0.1% working solution, followed by rinsing four times with distilled water. Samples were immersed in a solution of 25% glycerol for 30 min and stored until observed. Leaf segments were examined using Nikon epifluorescence equipment (Nikon, Tokyo, Japan), with a V-2 A filter (excitation filter 380–420 nm, barrier filter 430 nm).

Fungal development and associated plant responses were classified into the following developmental stages based on Soleiman et al. [46], with modifications adapted to *P. striiformis* according to Bozkurt et al. [11]: (i) spores developing a germinative tube without ending in a stoma were considered lost germinative tubes (LGT); (ii) spores developing a germinative tube ending in a stoma without exhibiting a substomatal vesicle (SSV) were considered germinative tubes reaching stoma (GTS); (iii) spores developing a SSV and a runner hypha (RHy), with no more than six haustorial mother cells (HMC), were considered early-aborted microcolonies without necrosis (EA−); (iv) spores developing a SSV and RHy, exhibiting notable plant cell death autofluorescence (hypersensitive response, HR) with no more than six HMC, were considered early-aborted microcolonies exhibiting necrosis (EA+); (v) spores developing a SSV and RHy with more than six HMC without necrosis were considered established microcolonies without necrosis (EST−); (vi) spores developing an established microcolony exhibiting notable HR were considered established microcolonies with necrosis (EST+). A total of 300 spores in four leaves per accession and replication were evaluated and classified according to their stage of development. Only germinated spores were counted. These fungal developmental stages were photographed using a Nikon DS-Fi1 camera (Nikon, Tokyo, Japan). In addition, 40 established microcolonies in four leaves per accession and replication were measured to their maximum colony length, parallel to the length of the leaves, to analyze the size of endophytically growing infection structures according to Moldenhauer et al. [45].

*2.5. Statistical Analysis*

The experimental design was developed as randomized blocks. In macroscopic experiments, data from the total affected area were transformed using the formula $y = \sqrt{(x)}$ and then analyzed using ANOVA and LSD (Least Significant Difference) tests, as were the other studied macroscopic parameters, except for the latency period, which was analyzed using the Kruskal–Wallis test. These analyses included cultivars (considering treatment) and replications (four plants) for each macroscopic parameter examined in two experiments (considered as different blocks). Similarly, for microscopy experiments, percentages of GTS and EA+ were transformed according to the formula $y = \sqrt{(x)}$, and back-transformed for presentation. Percentages of EST− and EST+ for colony length analysis were transformed according to the formula $y = \arcsin(\sqrt{(x/100)})$. Microscopic analyses also included cultivars (considering treatment) and replicates (300 spores and 40 established microcolonies in four leaves) from three experiments (considered as different blocks). All microscopic data were analyzed using ANOVA and LSD tests. Data processing, statistical analyses and figure design were carried out using R software (RStudio 3.6.1, R Core Team, Vienna, Austria) [50] and Fiji (Wayne Rasband, NIH, MD, USA) [38].

## 3. Results and Discussion

### 3.1. Resistance and Susceptibility Responses to P. striiformis Infection amongst Bread Wheat Accessions

Our study classified 19 Spanish bread wheat accessions as either resistant or susceptible to yellow rust infection according to their percentage of disease severity (DS) and disease scoring scale of infection type (IT) (Figure 1). Among them, 5 out of 19 (26%) accessions were considered susceptible, developing abundant sporulation with moderate chlorosis (IT 7) to abundant and dense sporulation without notable chlorosis or necrosis (IT 8–9). However, two of them, 'Nogal' (IT 9) and 'Califa Sur' (IT 8), expressed the most severe symptoms. The susceptible response observed in 'Nogal' is consistent with field data collected by the GENVCE (Spanish Group for Screening Field Crops New Varieties) network in 2019 and 2020 [26]. The other 14 evaluated accessions (74%) showed either partially resistant or resistant responses. Specifically, seven genotypes were considered partially resistant, exhibiting chlorotic and necrotic areas with limited to moderate development of pustules (IT 3–6), whereas seven accessions were considered highly resistant (IT 0–2) with no development of pustules, highlighting their role as sources of resistance in future breeding programs. In fact, it is significant that 14 out 19 accessions evaluated against *P. striiformis* showed various degrees of resistant responses under optimum conditions of infection, indicating that yellow rust resistance genes present in the genetic pool of Spanish bread wheat cultivars possibly have not yet been overcome [28]. These results contrast with those recently reported in Spanish landraces [51], where only 7% of the screened accessions, inoculated with a yellow rust isolate with a similar virulence/avirulence spectrum to the one used in this study, displayed resistant IT.

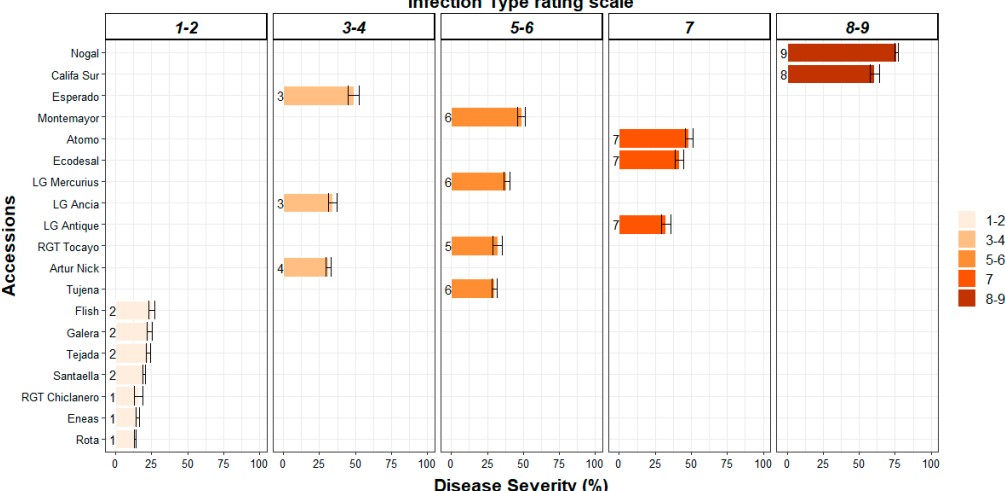

**Figure 1.** *P. striiformis* infection in bread wheat accessions. Mean percentage of disease severity (DS), presented in columns, and infection type (IT) rating scale, presented as numbers at the top of the figure. Accessions were arranged according to their mean percentage of DS and classified according to IT in panels. The IT scale is presented according to McNeal et al. [49], where 0 = no visible disease symptoms (immune), 1 = minor chlorotic and necrotic flecks, 2 = chlorotic and necrotic flecks without sporulation, 3–4 = chlorotic and necrotic areas with limited sporulation, 5–6 = chlorotic and necrotic areas with moderate sporulation, 7 = abundant sporulation with moderate chlorosis, 8–9 = abundant and dense sporulation without notable chlorosis and necrosis. Error bars represent the standard error calculated from three independent experiments with six replicates each.

The cultivar 'Nogal' showed the highest mean DS value among the studied accessions (76%), followed by 'Califa Sur' (61%), both classified as susceptible with IT 9 and 8, respectively. It is worth highlighting that both accessions, presenting the highest DS values, also scored the highest IT, suggesting elevated colonization of the fungus through leaves together with increased sporulation. 'Atomo', 'Ecodesal' and 'LG Antique' expressed on average DS values of 48, 42 and 32%, respectively, and were also classified as susceptible

accessions displaying IT 7. Regarding accessions which presented moderately resistant responses to *P. striiformis* infection due to moderate sporulation, 'Montemayor' (IT 6) developed, on average, a DS value of 49%, followed by 'LG Mercurius' (DS 39%, IT 6), 'RGT Tocayo' (DS 32%, IT 5) and 'Tujena' (DS 30%, IT 6). In addition, 'Esperado' (IT 3), 'LG Ancia' (IT 4) and 'Artur Nick' (IT 4) accessions were also considered moderately resistant, but with limited sporulation of *P. striiformis*, developing DS values of 49, 34 and 31%, respectively. Remarkably, 'Esperado' accession, despite having the third highest DS value, showed a quite low IT value (3). Finally, among the accessions considered resistant, 'Flish' presented the highest mean value of DS (25%), followed by 'Galera' (24%), 'Tejada' (23%), and 'Santaella' (20%), all categorized as IT 2. Only three accessions exhibited IT 1 values, 'RGT Chiclanero', 'Eneas' and 'Rota', developing, on average, DS values of 16%, 15% and 14%, respectively. Taking into account all the results shown here, it can be stated that our visual evaluation permitted us to classify of yellow rust infection through diverse parameters: (i) the fungal colonization of leaves (DS), which ultimately causes desiccation of leaves and plant stunting [52], and (ii) the IT scale for classification of resistant responses in the form of chlorotic and necrotic areas, and the sporulation capability of the fungus, leading to infection of other leaves, plants or cropping areas [5,13]. Although visual evaluation of DS or IT is a subjective and laborious assessment that can often be inaccurate [53], it is still considered a useful and quick first approach to determine disease extent and plant–pathogen interaction on leaves and plants for breeders, farmers and technicians.

Six accessions with interesting responses to yellow rust, from highly resistant to susceptible behavior, were selected to macro- and microscopically characterize resistance to this pathogen. The accessions were 'Rota', 'Galera', 'Artur Nick', 'Ecodesal', 'Califa Sur' and 'Nogal'.

### 3.2. Macroscopic Components of Resistance to P. striiformis Infection

These six accessions were macroscopically evaluated to quantitatively identify components of resistance to yellow rust infection. Figure 2 shows their IT scores. We evaluated diverse components of *P. striiformis* infection through image analysis, such as infection frequency (IF = pustules/cm$^2$), mean pustule size, total pustule area, total affected area, and proportion of pustule area relative to total disease area (Table 1).

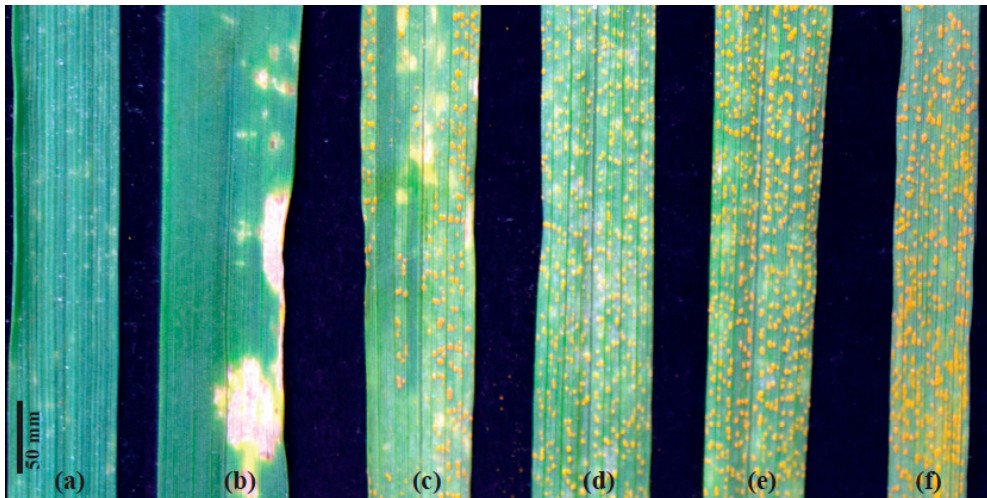

**Figure 2.** Examples of leaves infected with *P. striiformis* showing diverse IT scores. Leaves from accessions (**a**) 'Rota' (IT 1), (**b**) 'Galera' (IT 2), (**c**) 'Artur Nick' (IT 4), (**d**) 'Ecodesal' (IT 7), (**e**) 'Califa Sur' (IT 8) and (**f**) 'Nogal' (IT 9).

**Table 1.** Macroscopic image analysis of *P. striiformis* infection in six selected bread wheat accessions [1].

| Accession | Pustules/cm$^2$ | Mean Pustule Size (cm$^2 \times 10^{-3}$) | Total Pustule Area (%) | Total Affected Area (%) | Pustule Area/ Disease Area (%) |
|---|---|---|---|---|---|
| Nogal (IT 9) [2] | 230.85 ± 9.50 a | 0.49 ± 0.03 a | 11.14 ± 0.57 a | 1.64 (1.10 ± 0.25) c | 88.37 ± 4.10 a |
| Califa Sur (IT 8) | 205.27 ± 8.22 a | 0.48 ± 0.01 a | 9.91 ± 0.49 a | 5.98 (2.39 ± 0.20) b | 63.21 ± 4.58 b |
| Ecodesal (IT 7) | 153.44 ± 9.27 b | 0.46 ± 0.01 a | 7.00 ± 0.52 b | 2.92 (1.60 ± 0.23) c | 73.40 ± 5.34 b |
| Artur Nick (IT 4) | 42.27 ± 8.93 c | 0.39 ± 0.03 b | 1.70 ± 0.45 c | 24.02 (4.79 ± 0.39) a | 7.08 ± 1.71 c |
| Galera (IT 2) | - | - | - | 8.22 (2.75 ± 0.31) b | - |
| Rota (IT 1) | - | - | - | 1.51 (1.17 ± 0.14) c | - |

[1] Values are mean ± standard error for four leaves evaluated in two independent experiments. Transformed data ± standard error are shown in parenthesis. Data with the same letter within a column are not significantly different (LSD, $p < 0.05$). Dash (-) means no data were measured since there was no pustule development. [2] Infection type (IT) data was scored by visual analysis.

The susceptible cultivars 'Nogal' (IT 9) and 'Califa Sur' (IT 8) showed a fully compatible reaction to the yellow rust and displayed the highest mean IF values with 230 and 205 pustules/cm$^2$, respectively. Cultivar 'Ecodesal' (IT 7) showed an intermediate reaction with 153 pustules/cm$^2$, while partially resistant accession 'Artur Nick' (IT 4) displayed the lowest IF with 42 pustules/cm$^2$. In the particular case of the 'Ecodesal' accession, it developed sporulation ranging from moderate in some cases to abundant in others, without exhibiting notable chlorosis or necrosis, making its IT classification with the McNeal scale more difficult [49] through initial visual assessment. We finally scored 'Ecodesal' as IT 7, focusing on the sporulation capability of *P. striiformis* in this cultivar according to macroscopic evaluation through image analysis, which permitted an objective classification in comparison with visual assessment [54]. This measurement subsequently led us to quantitatively separate accessions based on IF, which was more accurate than visual assessment of DS to estimate rust infection [35,39], and also helped to more precisely define the initial IT scores obtained by visual assessment. Additionally, since 'Galera' (IT 2) and 'Rota' (IT 1) accessions were found to be highly resistant to *P. striiformis*, pustules were not developed and no measurements of IF or mean pustule size could be taken.

Moreover, average values of mean pustule size exhibited statistical differences between the susceptible accessions, 'Nogal' ($0.49 \times 10^{-3}$ cm$^2$), 'Califa Sur' (0.48) and 'Ecodesal' (0.46), and the less susceptible 'Artur Nick', which developed the smallest pustules ($0.39 \times 10^{-3}$ cm$^2$), possibly due to its partially resistant phenotype (IT 4). We also analyzed the total pustule area as a proportion of leaf area, finding similar differences as in the IF parameter. 'Nogal' expressed the highest mean value with 11%, followed by 'Califa Sur' (9.9%). A lower value was observed in 'Ecodesal' (7%), and the lowest value was recorded in 'Artur Nick' (1.7%). 'Galera' and 'Rota' accessions did not show any data since pustules were not developed. At this point, it is important to highlight that the count of pustules in *P. striiformis* infection presented some difficulties regarding the identification of single pustules due to pustule coalescence, and manual curation of the data was needed, especially in very susceptible accessions. For that reason, the measurement of total pustule area (as a percentage of the leaf area analyzed), which did not indicate the number of pustules, required a less complex process of analysis in comparison with IF, and yielded a similar classification of accessions, making it a preferable parameter to take into consideration for quantifying *P. striiformis* sporulation in image analysis applications [39].

Image analysis also permitted measurement of the total affected area (chlorosis and necrosis) without sporulation of *P. striiformis*. This parameter could be used to identify partially resistant accessions such as 'Artur Nick' (IT 4), which expressed the highest mean value (24%) and was statistically different from the rest of the accessions categorized as susceptible or resistant. This quantitative identification of chlorosis and necrosis also distinguished accessions with different resistant responses, such as 'Galera' (8.2%) with IT 2 and 'Rota' (1.51%) with IT 1, as well as accessions with different susceptible responses such as 'Nogal' (1.64%) and 'Ecodesal' (2.92%), which expressed smaller affected areas compared to 'Califa Sur' (5.9%). However, statistical analysis in this parameter among accessions did

not differentiate, for example, 'Nogal' or 'Califa Sur' (susceptible phenotypes, IT 9 and 8, respectively) from 'Rota' and 'Galera' (resistant phenotypes, IT 1 and 2, respectively), and it was necessary to take into consideration the proportion of pustule area relative to total disease area. This parameter helped us to understand what quantity of yellow rust infection produced sporulation. Therefore, this measurement led to statistical differences in sporulation development between the susceptible accessions 'Nogal' and 'Califa Sur' (88.37% and 63.21%, respectively), compared to the resistant ones 'Rota' and 'Galera' (no data reported since pustules were not developed).

Furthermore, the progression of yellow rust infection produced by our local isolate was evaluated in four of the six selected accessions where the pustules appeared well developed and sporulation could be observed. The macroscopic parameters latency period (LP0) and final lesion length were measured (Table 2). LP0 expressed differences between 'Nogal' (IT 9) and 'Califa Sur' (IT 8) (264 h on average) and 'Ecodesal' (IT 7) (303.75 h), all of which were considered susceptible accessions. Surprisingly, 'Ecodesal' developed a statistically longer latency period than 'Nogal' and 'Califa Sur', which could explain its macroscopic classification as IT 7, with moderate sporulation in some cases and abundant in others. Finally, the longest LP0 was recorded in 'Artur Nick' (306 h on average). The longer latency period developed by the 'Artur Nick' accession, together with a lower mean pustule size value than found in susceptible accessions, suggest a partial resistance behavior [46,55]. Regarding final lesion length, the shorter the latency period, the longer the final lesion length observed and vice versa. Thus, 'Califa Sur' and 'Nogal' developed the largest mean values of final lesion length, 63.69 and 60.58 mm respectively, showing statistical differences from 'Ecodesal' (43 mm) and 'Arthur Nick' (44 mm). These results revealed that the progression of yellow rust infection was more intense in 'Nogal' and 'Califa Sur' due to shorter latency periods in comparison with 'Ecodesal' and 'Artur Nick', which ultimately developed bigger final lesions. The resistant accessions, 'Rota' and 'Galera', did not produce pustules and no measurements were available.

**Table 2.** Macroscopic components of disease progression of *P. striiformis* infection in four selected bread wheat accessions [1].

| Accession | LP0 (hours) | Final Lesion Length (mm) |
|---|---|---|
| Nogal | 264 $\pm$ 0.00 a | 60.58 $\pm$ 3.05 a |
| Califa Sur | 264 $\pm$ 0.00 a | 63.63 $\pm$ 2.47 a |
| Ecodesal | 303.75 $\pm$ 8.56 b | 42.89 $\pm$ 4.59 b |
| Artur Nick | 306 $\pm$ 10.88 b | 44.18 $\pm$ 6.34 b |

[1] Values are mean $\pm$ standard error for four leaves evaluated in two independent experiments. Data with the same letter within a column are not significantly different (Kruskal–Wallis and LSD tests, $p < 0.05$).

### 3.3. Microscopic Components of Resistance to P. striiformis Infection

In addition to the macroscopic analyses, microscopic evaluation was carried out in order to compare micro- and macroscopic phenotypes, similarly to other studies developed for yellow rust [11,48], leaf rust [56] and stem rust [57]. The accessions used to evaluate macroscopic components were also examined for microscopic components of resistance to *P. striiformis* infection. Different fungal stages were observed (LGT, GTS, EA+, EST− and EST+) (Figure 3), and the established microcolonies that were extended through the leaves were measured.

First of all, measurements of spores which developed lost germinative tubes (LGT) presented similar mean values in the 'Nogal' (34%), 'Galera' (35%), 'Rota' (36%) and 'Califa Sur' (39%) accessions, whereas 'Artur Nick' and 'Ecodesal' had values of 48 and 58%, respectively (Figure 4). Additionally, mean values of spores which developed germinative tubes reaching stoma (GTS) were higher in the 'Rota' (17%), 'Artur Nick' (15%) and 'Ecodesal' (12%) accessions. The rest of the accessions showed lower mean values with 'Califa Sur' expressing 8%, 'Galera' 7% and 'Nogal' 6%. Thus, both of these pre-infection parameters (LGT and GTS as a set) together expressed mean percentage values from 40% in 'Nogal' to 70%

in 'Ecodesal', in concordance with studies developed by de Vallavieille-Pope et al. [58,59], which found a reduction in *P. striiformis* penetration efficiency in comparison with other wheat rusts, such as *P. triticina*. Accessions 'Ecodesal' and 'Artur Nick' presented elevated values of LGT plus GTS, which could explain their lower IF relative to the susceptible accessions as a consequence of a higher proportion of spores that did not form infection sites (70% and 63%, respectively). These low penetration values in 'Ecodesal' and 'Artur Nick' could be explained by a multifactorial component, where in addition to the reduced *P. striiformis* penetration efficiency, diverse circumstances related to the leaf surface played key roles. Leaf surface is the first physical and chemical barrier against fungal infection [60], and elements such as density, wax and morphology of leaf trichomes and stomata [61–63], along with the expression of antifungal compounds in trichomes [64], are actively involved in halting fungal infection. In this context, 'Rota' expressed the highest mean value of GTS, although without statistical differences with 'Ecodesal' and 'Artur Nick', suggesting the presence of an obstacle mentioned above to the *P. striiformis* penetration process, which needs to be determined via further research.

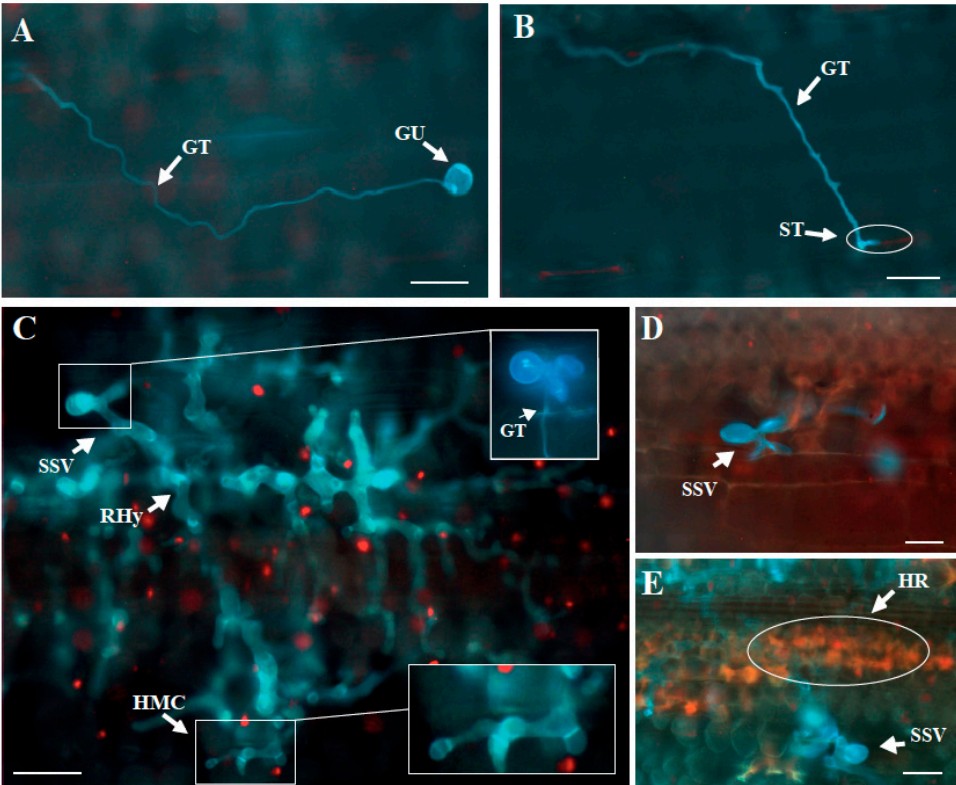

**Figure 3.** Microscopic fungal stages of *P. striiformis* and plant cellular responses were classified as: (**A**) lost germinative tube (LGT), showing the germinated urediniospore (GU) and the germinative tube (GT); (**B**) germinative tube reaching stoma (GTS), showing a germinative tube ending in a stoma (ST) without establishing a substomatal vesicle (SSV); (**C**) established microcolony without necrosis (EST−), with a detailed SSV, developing runner hyphae (RHy) and haustorial mother cells (HMC); (**D**) early-aborted colony associated with hypersensitive response (EA+); (**E**) established microcolony associated with necrosis (EST+), also developing SSV with RHy and HMC associated with hypersensitive response. HR, hypersensitive response. Scale, 50 μm.

Furthermore, spores which formed infection sites through substomatal vesicle development were counted and classified, showing clear differences among accessions. Spores that formed early-aborted microcolonies absent plant cell death response (EA−) were highly infrequent in all studied accessions, and were not analyzed in this study. Early-aborted microcolonies showing plant cell death autofluorescence (EA+) presented higher mean values in accessions considered resistant, such as 'Rota' (26%, IT 1) and 'Galera' (16%, IT 2). These

resistant accessions developed incompatible interactions with *P. striiformis* to some extent, similarly to other incompatible host–*Pst* interactions shown in other studies [9,11,45,47,65]. In fact, the EA+ mean value developed by 'Rota' suggests early recognition of the pathogen, triggering an intensive hypersensitive response (HR), which did not permit fungal growth, in comparison with 'Galera' (also resistant). This condition correlates with a different macroscopic phenotype with IT 1 in 'Rota' and IT 2 in 'Galera', similar to the results obtained by Saleem et al. [65]. However, the evidence for early pathogen recognition needs further analysis at different time points previous to seven days post-inoculation (dpi) to elucidate the evolution of host responses to yellow rust infection [47,65]. By contrast, partially resistant and susceptible accessions exhibited lower EA+ values, indicating that spores that caused infection sites finally developed, in general, established microcolonies. In particular, 'Califa Sur' (IT 8) showed a remarkably higher mean value of EA+ (8%) in comparison with 'Nogal' (0.5%; IT 9), which was also in concordance with macroscopic results (total affected area) and reinforced the link between macroscopic phenotype and histopathological parameters. The partially resistant accession 'Artur Nick' (IT 4) and the moderately susceptible 'Ecodesal' (IT 7) also displayed low EA+ percentages (5 and 3%, respectively).

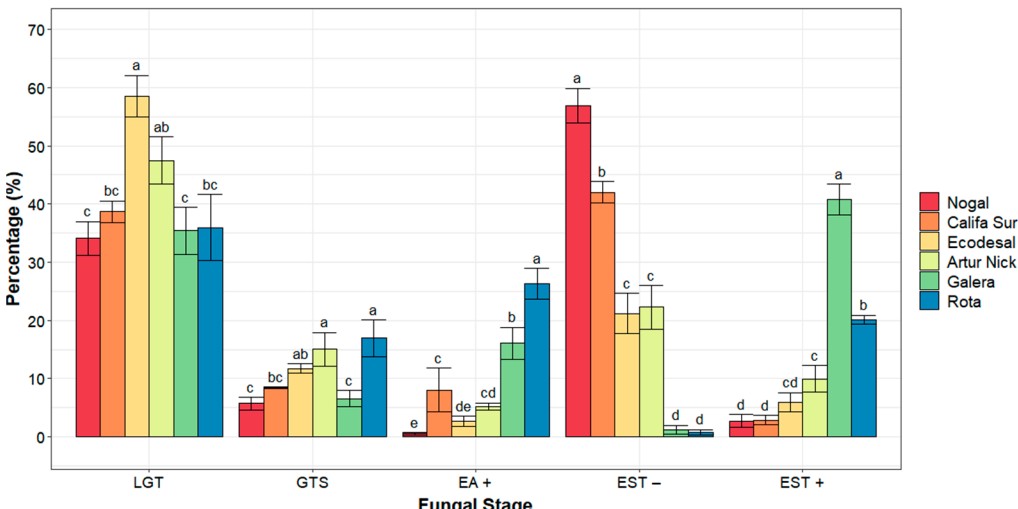

**Figure 4.** Microscopic fungal stages of *P. striiformis* in six selected bread wheat accessions presented as mean percentages. Error bars represent the standard error calculated from three independent experiments. Data with the same letter within a fungal stage are not significantly different (LSD test, $p < 0.05$). LGT, lost germinated tubes; GTS, germinate tubes reaching stoma; EA+, early-aborted microcolonies with necrosis; EST−, established microcolonies without necrosis; EST+, established microcolonies with necrosis.

Regarding mean values of spores which formed established microcolonies without plant cell death (EST−), susceptible accessions 'Nogal' (57%) and 'Califa Sur' (42%) showed statistical differences corresponding with their macroscopic differences obtained through visual IT (9 and 8, respectively) and total affected area (1.6% and 6%, respectively). IF and total pustule area did not show differences. 'Ecodesal', also classified as susceptible, developed significantly lower mean values of EST− (21%) compared to 'Nogal' and 'Califa Sur', possibly due to the higher LGT and GTS values observed in this accession. However, this difference in EST- values relative to 'Nogal' and 'Califa Sur' was in the form of macroscopic results related to sporulation parameters (IF and total pustule area). Similarly, 'Artur Nick', considered partially resistant, did not express differences in EST− values (22%) compared to 'Ecodesal', despite the clear macroscopic differences in IF, total pustule area and total affected area. Thus, this microscopic parameter (EST−) could be considered helpful for classifying susceptible genotypes which presented differences at the microscopic level but were not clearly correlated in macroscopic measurements, and likewise for the microscopic

parameter of established microcolonies associated with plant cell death responses used for incompatible interactions [11,48]. As we expected, 'Galera' and 'Rota' exhibited the lowest values of EST− (1% in both accessions), manifesting macroscopic responses without development of pustules.

Finally, spores developing established microcolonies associated with HR (EST+) presented mean values that differentiated accessions statistically. 'Galera' presented the highest mean value with 41%, with EST+ being the principal microscopic component of its resistance. This value in 'Galera' is possibly due to an initial delay in the activation of HR against infection, which permitted fungal development until later stages, when HR then appeared in the majority of established microcolonies. This delayed activation of the HR response and the subsequent fungal growth could suggest an indirect recognition of the pathogen by the host, characterized by a macroscopic phenotype of IT 2, similar to that described by Saleem et al. [65] in a wheat near-isogenic line with *Yr6* resistance gene at 8 dpi. However, this supposed delay in the expression of HR needs further analysis at previous time points, as in the cases of early pathogen recognition in accessions with relevant values of EA+, such as 'Rota' [47,65]. In particular, 'Rota' also presented a remarkable value of EST+ (20%), which could suggest that the initial expression of HR by the host, shown in the EA+ values, stopped fungal development prior the runner hyphae (RHy) formation in some cases, whereas the fungus was capable of developing RHy and at least six haustorial mother cells before being enclosed by HR in other cases.

The accessions that eventually developed pustules in their macroscopic IT also expressed lower values of EST+ in comparison with 'Rota' and 'Galera', leading in the end to lower HR at a microscopic level. Thus, 'Artur Nick', as a partially resistant accession, expressed a mean value of 10% of EST+, statistically different from susceptible accessions 'Ecodesal' (6%), 'Califa Sur' (3%) and 'Nogal' (2.5%), correlating with its macroscopic results. Although 'Ecodesal' was considered susceptible, it showed a different EST+ value in comparison with 'Nogal' and 'Califa Sur', possibly due to a slight expression of HR. Because of that, this accession could be considered moderately susceptible. Values of EST+ were not statistically different between susceptible accessions. In contrast, they could be used to differentiate susceptible from partially resistant accessions with incompatible interactions (IT 3–6) such as 'Artur Nick' and, most relevantly, EST+ could classify accessions considered resistant (IT 1–2) with statistical differences in the presence of HR in their established microcolonies [11,48].

Once we completed the identification and quantification of the main fungal stages developed by *P. striiformis* in our studied accessions, we performed measurements of colony length in their established microcolonies in order to quantify the extension of the fungus through leaf tissue, taking into account the presence of HR (Figure 5), similar to previous studies [9,45].

Data presented in Table 3 show analysis of the colony length of established *P. striiformis* microcolonies. 'Nogal' and 'Califa Sur', both susceptible, expressed the highest mean values of colony length (868 and 822 μm respectively) and the highest EST− values amongst our studied accessions (96 and 94%, respectively), in accord with their macroscopic IT scale values (9 and 8, respectively) and their sporulation parameters. Their reduced proportions of established microcolonies that developed HR (EST+, 4 and 6%, respectively) indicated that fungus could grow extensively in these accessions. These results together, both high colony length values and diminished number of microcolonies with HR, could serve as factors for differentiating susceptible accessions from others which developed reduced infection severity, such as 'Ecodesal' (IT 7) with a shorter colony length (646 μm) and a higher percentage of established microcolonies with HR (23%). However, 'Ecodesal' did not show relevant macroscopic affected areas with chlorosis or necrosis, suggesting that these HR were weakly presented, allowing the extension of fungal development. This fact, together with the statistical similarity of 'Ecodesal' (IT 7) to 'Artur Nick' (IT 4) in the parameters of colony length (538 μm) and proportion of established microcolonies associated with HR (31%), could indicate a quantitative increase (current or subsequent)

in HR development in 'Artur Nick' compared with 'Ecodesal', which corresponded with their final developed macroscopic phenotype. Thus, since in our study the evaluation of HR was conducted qualitatively, we could not establish the degree of expression of these responses. For that reason, a quantitative characterization of these parameters in future studies could shed some light on the differences between accessions that developed different degrees of incompatible interactions with yellow rust, similar to the studies developed by Saleem et al. [65], where the percentage of microcolonies covered with autofluorescence responses in genotypes with diverse *Yr* genes was measured.

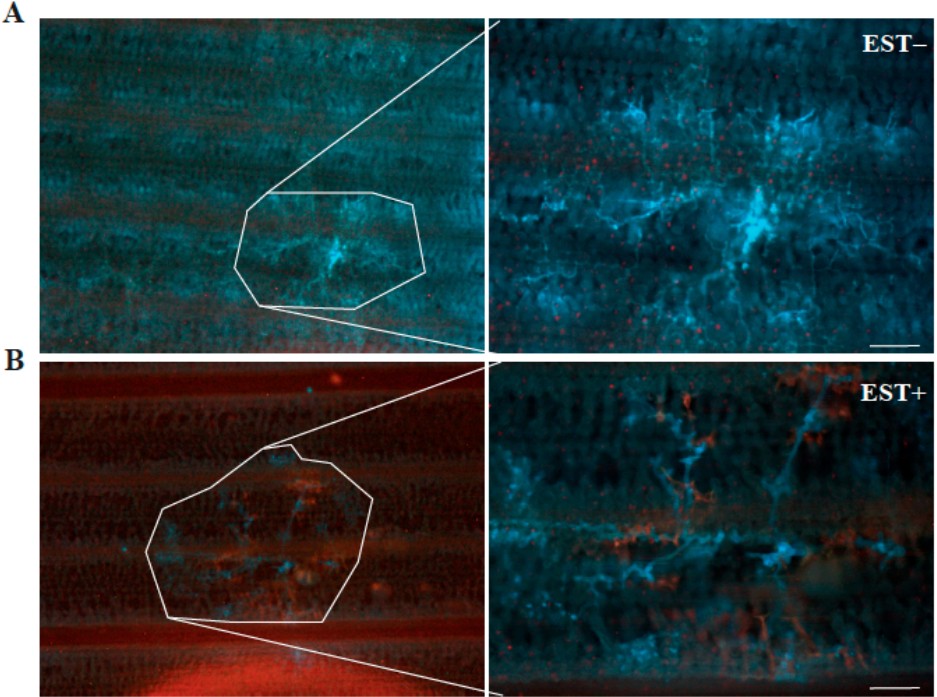

**Figure 5.** (**A**) Examples of established *P. striiformis* microcolonies without necrosis (EST−, upper panel) and (**B**) with necrosis (EST+, lower panel). Left panels show 4 × images with delimited microcolonies magnified at 10 × in the right panels. Scale, 100 μm.

**Table 3.** Analysis of the colony length of established *P. striiformis* microcolonies and their distribution relative to total observed established microcolonies in six selected bread wheat accessions [1].

| Accession | Colony Length (μm) | % EST− | % EST+ |
|---|---|---|---|
| Nogal | 867.92 ± 34.31 a | 95.61 (0.20 ± 0.04) a | 4.39 (1.37 ± 0.04) c |
| Califa Sur | 822.26 ± 18.85 ab | 93.73 (0.25 ± 0.03) a | 6.27 (1.32 ± 0.03) c |
| Ecodesal | 646.61 ± 58.45 cd | 77.07 (0.49 ± 0.10) b | 22.93 (1.08 ± 0.10) b |
| Artur Nick | 537.86 ± 38.51 d | 68.42 (0.59 ± 0.09) b | 31.58 (0.98 ± 0.09) b |
| Galera | 739.33 ± 44.93 bc | 2.91 (1.43 ± 0.07) c | 97.09 (0.14 ± 0.07) a |
| Rota | 368.99 ± 19.33 e | 3.68 (1.42 ± 0.08) c | 96.32 (0.15 ± 0.08) a |

[1] Values are mean ± standard error for four leaves evaluated in three independent experiments. Transformed data ± standard error are shown in parentheses. Data with the same letter within a column are not significantly different (LSD, $p < 0.05$).

On the other hand, the hypersensitive accession 'Rota' (IT 1), which developed macroscopically small chlorotic and necrotic flecks, exhibited the lowest mean colony length value (369 μm), statistically different from the other accessions, and a remarkably elevated percentage of EST+ microcolonies (96%). These results suggest an early inhibition of fungal growth by the host during the infection process associated with HR, similar to previous histological studies developed by Kang et al. [66], where the low IT scores expressed by *Pst*-resistant wheat cultivars were related to the inhibition of fungal growth. Finally, although

'Galera' (IT 2) showed colony length values statistically similar to susceptible accessions (739 μm), it developed the highest percentage of EST+ microcolonies (97%), thus showing differences not only from susceptible accessions, but also from the resistant accession 'Rota'. These results suggest that 'Galera' displayed delayed HR, thus leading to higher fungal development compared to the faster defence pattern of the 'Rota' accession. In addition, future studies focusing on the variation in colony size and HR as quantitative values (percentage) at different time points during the infection process could more precisely characterize the microscopic phenotypes of accessions resistant to yellow rust, such as 'Galera'. In this context, other studied parameters extending the knowledge of the *Pst*–wheat interaction at the microscopic level would be the accounting of HMC per infection unit [67] or the deposition of compounds such as lignin or callose [9,45]. In summary, the study of diverse microscopic components of resistance to *P. striiformis* provide relevant information that correlates with the macroscopic expression of the disease, leading to an improvement in the understanding of plant–pathogen interaction for future breeding purposes, complementing the predominant genetic, biochemical and physiological studies [47].

## 4. Conclusions

To the best of our knowledge, this study is the first macroscopic and microscopic characterization of *P. striiformis* infection in a collection of Spanish bread wheat cultivars under controlled conditions. Using visual evaluation to analyze parameters such as DS and IT, the cultivars were classified in a range from susceptible to highly resistant depending on their disease expression patterns. Then, several macro- and microscopic parameters of resistance were defined, resulting in measurements that differentiated our studied cultivars quantitatively. Thus, such differences could be considered of particular relevance to find and characterize resistant responses against yellow rust. Moreover, this could help us gain a better knowledge of plant–pathogen interactions, especially considering the quick adaptation of *P. striiformis* to warmer climates. Despite the progress made in this work, further research should focus on the study of early *P. striiformis* recognition by the resistant accessions, prior to 7 dpi, together with physiological study of the leaf surfaces of accessions with low *P. striiformis* penetration values in order to elucidate the evolution of host responses to and morphological changes in yellow rust infection. The continuous evolution of *P. striiformis* allows it to spread into diverse territories and cultivars in Spain, requiring more detailed evaluation of cultivars' responses in order to control them and adapt new breeding cultivars through genetic, biochemical and physiological studies.

In conclusion, our study represents an important new step in the research of plant–pathogen interactions for future breeding purposes regarding the new *P. striiformis* threat in the Spanish cultivars, reinforcing the link between macroscopic phenotypes and histopathological parameters.

**Supplementary Materials:** The following supporting information can be downloaded at: https://www.mdpi.com/article/10.3390/agronomy12051239/s1, Figure S1: *P. striiformis* infection in adult plants in bread wheat accessions. Mean percentage of disease severity (DS), presented in columns, and infection type (IT) rating scale, presented as numbers at the top of the figure. Accessions arranged according to their mean percentage of DS and classified by IT in panels. IT scale is presented according to McNeal et al. [49], where 0 = no visible disease symptoms (immune), 1 = minor chlorotic and necrotic flecks, 2 = chlorotic and necrotic flecks without sporulation, 3–4 = chlorotic and necrotic areas with limited sporulation, 5–6 = chlorotic and necrotic areas with moderate sporulation, 7 = abundant sporulation with moderate chlorosis, 8–9 = abundant and dense sporulation without notable chlorosis and necrosis. Error bars represent the standard error calculated from two independent experiments with six replicates each; Table S1: Commercial cultivars used in this work and their source of origin.

**Author Contributions:** Conceptualization, supervision, project administration, funding acquisition, resources, J.C.S. and A.P.-d.-L.; investigation, R.P. and C.M.-R.; methodology, formal analysis, writing—original draft preparation, data curation, visualization, R.P.; writing—review and editing, validation, J.C.S., A.P.-d.-L. and C.M.-R. All authors have read and agreed to the published version of the manuscript.

**Funding:** This research was funded by the Projects AVA.AVA2019.020, RTA2015-0072-C03-02 and PID2020-118650RR-C32 (all cofunded by European Regional Development Fund).

**Institutional Review Board Statement:** Not applicable.

**Informed Consent Statement:** Not applicable.

**Data Availability Statement:** Not applicable.

**Acknowledgments:** The author acknowledges the European Social Fund (European Union) and the Agencia Estatal de Investigación (Spain) for PhD grant number BES-C2017-0091.

**Conflicts of Interest:** The authors declare no conflict of interest.

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
