# Peer review of "Macro- and Microscopic Characterization of Components of Resistance against Puccinia striiformis f. sp. tritici in a Collection of Spanish Bread Wheat Cultivars"

_agronomy, doi:10.3390/agronomy12051239_

Round 1
Reviewer 1 Report
The results of evaluation of resistance to yellow rust in 19 Spanish bread wheat cultivars are discussed. The material is presented on 20 pages. The description of results is lengthy and chaotic. The title of the manuscript is inconsistent with the aim of studies. It is not clear what the authors were studied: resistance of varieties (specific, nonspecific or prehaustorial) or characterization of Puccinia striiformis f. sp. tritic infection on cultivars with different resistance types. The terms macro and microscopic characteristics are not correct. Authors wrote that the main objective of study was to evaluate a group of Spanish bread wheat cultivars against a local isolate of P. striiformis under climatic controlled conditions through both, macroscopic and microscopic methods.
Author Response
The results of evaluation of resistance to yellow rust in 19 Spanish bread wheat cultivars are discussed. The material is presented on 20 pages. The description of results is lengthy and chaotic. The title of the manuscript is inconsistent with the aim of studies. It is not clear what the authors were studied: resistance of varieties (specific, nonspecific or prehaustorial) or characterization of Puccinia striiformis f. sp. tritic infection on cultivars with different resistance types. The terms macro and microscopic characteristics are not correct. Authors wrote that the main objective of study was to evaluate a group of Spanish bread wheat cultivars against a local isolate of P. striiformis under climatic controlled conditions through both, macroscopic and microscopic methods.
Response: The comments from reviewer #1 are vague and imprecise. Neither the Editor nor the other reviewers found the results lengthy or chaotic. If that were the case, we would expect at least specific (constructive) comments about where the description of results is confusing and unclear. We never used the term “macro and microscopic characteristics”. In the tittle and throughout the text we refer to “macro and microscopic characterization (of infection or response to infection)”, which has a different meaning. It is a common term used in the literature for studying and describing the different components of the resistance, being it complete or partial (which includes specific, prehaustorial and posthaustorial mechanisms), through several parameters measured at the macro and micro scale (using macroscopic and microscopic methods). And we did that in a collection of Spanish wheat cultivars. For that reason, we think the tittle is consistent with the aim of the study. Nevertheless, the title has been changed for a new one highlighting the terms “components of resistance”. The new one is: “Macro and microscopic characterization of components of resistance against Puccinia striiformis f. sp. tritici in a collection of Spanish bread wheat cultivars.
Reviewer 2 Report
The manuscript by Porras et al. reported the screening of stripe rust resistance from a set of Spanish bread wheat cultivars and macro- and microscopic characterization of their reaction to Pst infection. They are trying to establish an image analysis system in correlation to the regular Pst screening assay using disease severity and infection type. The research results are valuable for the society to screen large amount of wheat germplasm precisely using imaging analysis and artificial intelligence processing system for phenotyping. They also trying to establish the relationship of resistance and susceptible genotypes with the initial Pst invasion process microscopic data which is also very important for understanding the resistance/susceptible mechanism of host-rust interaction.
Suggestions for the revision:
- In Table 1, it would be better to include the DS and IT data for the genotypes tested for a clear comparison of the results.
- The study was conducted at the seedling stage which is often referring as all stage resistance (ASR). Please add some discussion of the potential value of these analysis for the adult plant resistance (APR).
- I am not familiar with the wheat growing situation at Andalusian cropping area. Why only 19, but not more, Spanish bread wheat cultivars were tested?
Author Response
1. In Table 1, it would be better to include the DS and IT data for the genotypes tested for a clear comparison of the results.
Response: We appreciate the suggestion, although we do not consider it appropriate to incorporate DS in Table 1, we do include IT data in Table 1. Our rationale is explained below:
DS is a disease parameter, which includes the entire infection (pustules, chlorosis, and necrosis) as a percentage of the leaf in a visual subjective evaluation, while these same infection parameters presented in Table 1 were obtained by image analysis, in an objective analysis. In addition, the timing of the DS data was different from the one in Table 1, so data would not be compared. However, the disease parameter of IT could be incorporated into Table 1 because is a qualitative parameter used to classify accessions regarding their general response to infection, which helps to understand the results and discussion.
2.- The study was conducted at the seedling stage, which is often referring as all stage resistance (ASR). Please add some discussion of the potential value of these analyses for the adult plant resistance (APR).
Response: In our study, we also developed an additional experiment to evaluate the response of our studied cultivars against yellow rust infection in an advanced growth stage of plants (2 months). Material and Methods have been updated with this information and supplemental figure 1 covering adult plant evaluation in terms of DS (disease severity) and IT (infection type) has been included.
3.- I am not familiar with the wheat-growing situation in the Andalusian cropping area. Why only 19, but not more, Spanish bread wheat cultivars were tested?
Response: Our study was not looking for resistance resources, instead was focused on identifying different responses in terms of sensitivity to yellow rust, and once we analyzed 19 varieties, we were able to cover the full spectrum of possibilities being able to perform a broad macro and microscopic characterization of them.
Reviewer 3 Report
The authors have presented interesting and valuable results on diverse resistance mechanisms in 19 economically important Spanish wheat varieties. I only have one comment, which is to fully describe the statistical analyses conducted. For example, what terms were included in the models (cultivar, replication, etc.)?
Author Response
The authors have presented interesting and valuable results on diverse resistance mechanisms in 19 economically important Spanish wheat varieties. I only have one comment, which is to fully describe the statistical analyses conducted. For example, what terms were included in the models (cultivar, replication, etc.)?
Response: In the statistical analysis of our different parameters evaluated, we performed ANOVA and LSD tests, with the exception of macroscopic parameter latency period, which was analyzed by Kruskal-Wallis test. These analyses included cultivars (considered treatment) and replications (4 plants) for each macroscopic parameter developed in two experiments (considered as different blocks). In the case of microscopic parameters, analyses also included cultivars (considered treatment) and replications (300 spores and 40 established microcolonies in 4 leaves) developed in three experiments (considered as different blocks). We have not performed any modelization study.
Round 2
Reviewer 1 Report
I could not recommend this article for publication. It is too many inaccuracies and incorrect discussions of the results.
Author Response
no comments